# Better Calibration Error Estimation for Reliable Uncertainty Quantification

Shuman Peng [* 1]  Parsa Alamzadeh [* 1]  Martin Ester [1]

## Abstract

Reliable uncertainty quantification is crucial in high-stakes applications, such as healthcare. The $ECE_{EW}$ has been the most commonly used estimator to quantify the calibration error (CE), but it is heavily biased and can significantly underestimate the true calibration error. While alternative estimators, such as $ECE_{DEBIASED}$ and $ECE_{SWEEP}$, achieve smaller estimation bias in comparison, they exhibit a trade-off between overestimation of the CE on uncalibrated models and underestimation on recalibrated models. To address this trade-off, we propose a new estimator based on K-Nearest Neighbors (KNN), called $ECE_{KNN}$, which constructs representative overlapping local neighbourhoods for improved CE estimation. Empirical evaluation results demonstrate that $ECE_{KNN}$ simultaneously achieves near-zero underestimation of the CE on uncalibrated models while also achieving lower degrees of overestimation on recalibrated models. The implementation of our proposed $ECE_{KNN}$ is available at https://github.com/esterlab/KNN-ECE/.

## 1. Introduction

Uncertainty quantification is crucial in high-stakes applications such as healthcare. The model should not only predict the outcome of a sample but also reliably communicate the prediction uncertainty. One common approach for communicating prediction uncertainty is to use the probability of the predicted class, also known as prediction confidence. Figure 1 illustrates the process of uncertainty quantification. *Reliable uncertainty quantification* requires (1) *calibrated models* that produce prediction confidence that match the prediction accuracy and (2) *evaluation metrics (estimators)* that can accurately estimate the *calibration error*. The calibration error is the expected difference between the pre-

*Equal contribution  [1]School of Computing Science, Simon Fraser University, Burnaby, Canada. Correspondence to: Shuman Peng <shumanp@sfu.ca>.

*Workshop on Interpretable ML in Healthcare at International Conference on Machine Learning (ICML)*, Honolulu, Hawaii, USA. 2023.

**Step 1:** Quantify the predictive uncertainty

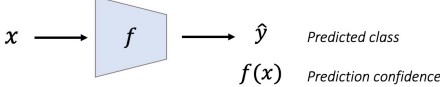

**Step 2:** Evaluate the predictive uncertainty

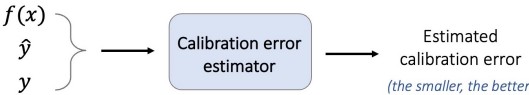

Figure 1: The uncertainty quantification process. First, a model (e.g., deep neural network) $f$ takes a sample $x$ as input and outputs the predicted class $\hat{y}$ along with the probability of the predicted class (i.e., prediction confidence) $f(x)$. Then, a calibration error estimator takes $f(x)$, $\hat{y}$, and $y$ (the true class of $x$) of a set of samples as input and outputs the estimated calibration error. In this work, we focus on the second step of the uncertainty quantification process (i.e., evaluating models' predictive uncertainty).

diction confidence and the accuracy. In this paper, we will use the terms "estimators" and "evaluation metrics" interchangeably.

Medical applications increasingly use deep neural networks (DNNs) for tasks such as medical diagnosis/prognosis (Sharifi-Noghabi et al., 2018), cancer treatment (Sharifi-Noghabi et al., 2020; Snow et al., 2021), and medical imaging analysis (Rajpurkar et al., 2017; Yao et al., 2021). However, DNNs often make over-confident and poorly calibrated predictions, resulting in highly confident incorrect predictions, which go undetected (Guo et al., 2017; Nixon et al., 2019). Many recalibration methods, such as Temperature Scaling, Deep Ensembles, and Monte Carlo Dropout, exist to improve DNN calibration to alleviate this problem (Guo et al., 2017; Lakshminarayanan et al., 2016; Gal & Ghahramani, 2015; Kull et al., 2019; Zhang et al., 2020; Wang et al., 2020). Once recalibrated, the DNNs require an assessment to determine their calibration error. The Expected Calibration Error (ECE)[1] based on equal-width binning ($ECE_{EW}$) is

[1]Also referred to as the *Estimated Calibration Error* in the existing literature.

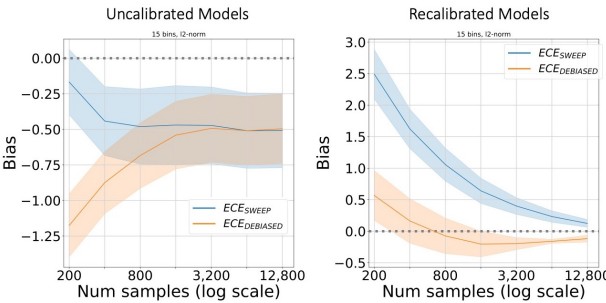

Figure 2: $ECE_{SWEEP}$ and $ECE_{DEBIASED}$ exhibit a tradeoff between underestimation of the calibration error (CE) on uncalibrated models (left) and overestimation on recalibrated models (right). Specifically, $ECE_{SWEEP}$ achieves lower underestimation on uncalibrated models but shows greater overestimation on recalibrated models, while the opposite holds true for $ECE_{DEBIASED}$. The x-axis represents varying sample sizes, while the y-axis represents the CE estimation bias, which measures the difference between the estimated CE and the true CE. Negative bias values indicate underestimations of the CE, and positive bias values indicate overestimations.

a widely used metric for this purpose, which partitions prediction confidence into $M$ equal-width bins and computes the mean discrepancy between the prediction confidence and empirical accuracy across the $M$ bins (Naeini et al., 2015; Guo et al., 2017). However, $ECE_{EW}$ is highly biased and can significantly underestimate the calibration error (Kumar et al., 2019; Nixon et al., 2019; Roelofs et al., 2022), posing a major concern for medical applications. Seemingly well-calibrated but poorly calibrated models can make incorrect predictions with high confidence, creating a false sense of certainty for users, which can lead to potentially fatal outcomes.

To address calibration error estimation bias, recent works introduced alternative calibration error estimators. The equal-mass binning-based ECE ($ECE_{EM}$) creates $M$ non-equally spaced bins containing an equal number of confidence scores (Nixon et al., 2019). The debiased ECE estimator ($ECE_{DEBIASED}$) subtracts an estimated bias correction term (Kumar et al., 2019), while the monotonic sweep ECE ($ECE_{SWEEP}$) adaptively determines the optimal number of bins (Roelofs et al., 2022).

$ECE_{SWEEP}$ and $ECE_{DEBIASED}$ have emerged as the top-performing estimators in previous studies (Roelofs et al., 2022). Specifically, when applied to the CIFAR-10, CIFAR-100, and ImageNet datasets using the equal-mass binning scheme, these estimators demonstrate the lowest mean absolute bias on uncalibrated models and models recalibrated using Temperature Scaling (Guo et al., 2017) – a widely

adopted recalibration method – respectively. However, both $ECE_{SWEEP}$ and $ECE_{DEBIASED}$ exhibit a trade-off between underestimation and overestimation of calibration errors on uncalibrated and recalibrated models (see Figure 2). While $ECE_{SWEEP}$ achieves a lower underestimation[2] of calibration error on uncalibrated models, it tends to overestimate[3] the error on recalibrated models. Conversely, $ECE_{DEBIASED}$ reduces overestimation on recalibrated models but increases underestimation on uncalibrated models.

In practical applications where the calibration level of a model is unknown, finding the right balance becomes critical. Underestimating the calibration error on uncalibrated models can lead to misplaced trust and potential risks, such as incorrect diagnoses or treatment decisions. Conversely, overestimating the error on recalibrated models can result in unnecessary caution, potentially causing delays in critical interventions or resource allocation.

In this paper, our goal is to address the tradeoff between the over and underestimation of calibration errors on uncalibrated and recalibrated models. To accomplish this, we propose a novel K-Nearest Neighbors (KNN)-based ECE estimator ($ECE_{KNN}$). This estimator constructs overlapping local neighborhoods, each containing $k$ prediction confidence scores, to estimate the deviation between the prediction confidence and empirical accuracy. By ensuring that each local neighborhood has a representative sample size, $ECE_{KNN}$ provides less biased estimations.

We evaluate the performance of our proposed KNN-based ECE ($ECE_{KNN}$) estimator alongside existing calibration error estimators ($ECE_{EW}$, $ECE_{DEBIASED}$, and $ECE_{SWEEP}$) on the same datasets (CIFAR-10, CIFAR-100, and ImageNet) and model backbones used in (Roelofs et al., 2022), across various sample sizes. Our estimator strikes a superior balance in estimation bias on both uncalibrated and recalibrated models compared to the best baseline estimators. Notably, our estimator exhibits significantly less underestimation of calibration error on uncalibrated models compared to the best baseline ($ECE_{SWEEP}$) while demonstrating less conservatism (by reducing overestimation) on recalibrated models. The implications of these experimental findings extend to healthcare applications, particularly in the context of DNNs trained on medical imaging tasks, such as skin cancer detection using the HAM10000 dataset (Tschandl et al., 2018). The confidence distributions produced by these medical imaging models, based on commonly used DNN backbones such as ResNet, closely resemble those trained on the CIFAR-10, CIFAR-100, and ImageNet datasets (as

---

[2]A more negative calibration error estimation bias indicates a more severe underestimation of calibration error, while a less negative bias implies a less severe underestimation.

[3]A larger positive bias indicates a more severe overestimation of calibration error.

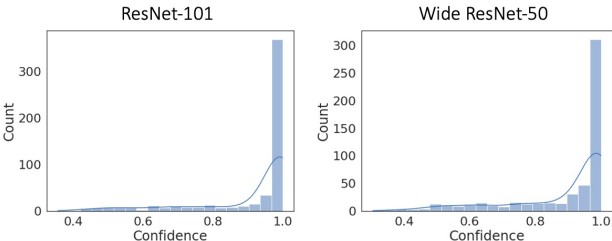

Figure 3: The confidence distribution of two ResNet models trained and tested on the Skin Cancer Detection HAM10000 dataset (Tschandl et al., 2018). The ResNet models were pre-trained on the ImageNet dataset and fine-tuned on the HAM10000 dataset, following the standard practice in medical image classification.

demonstrated in Figures 3 and 4).

Our paper contributes in the following ways:

- We identify a tradeoff observed among top-performing calibration error (CE) estimators. In particular, ECE$_{\text{SWEEP}}$ achieves a lower underestimation of CE on uncalibrated models, but at the expense of higher overestimation on recalibrated models. Conversely, ECE$_{\text{DEBIASED}}$ reduces overestimation on recalibrated models but suffers from more significant underestimation on uncalibrated models. This tradeoff highlights the need for a new estimator that strikes a better balance between over and underestimation on models with different calibration levels.

- To address this tradeoff, we propose a novel estimator, the KNN-based ECE (ECE$_{\text{KNN}}$). Leveraging the concept of K-Nearest Neighbors, ECE$_{\text{KNN}}$ constructs overlapping and representative local neighborhoods to estimate the calibration error more accurately.

- Through comprehensive experiments, we demonstrate that our ECE$_{\text{KNN}}$ estimator strikes a superior balance in terms of CE estimation bias on uncalibrated and recalibrated models. Specifically, our ECE$_{\text{KNN}}$ estimator significantly reduces the underestimation of CE on uncalibrated models while simultaneously reducing overestimation on recalibrated models. By striking a better balance between underestimation and overestimation, our ECE$_{\text{KNN}}$ estimator provides healthcare professionals with more reliable uncertainty quantification, promoting informed decision-making and reducing the risk of false confidence or unnecessary caution.

## 2. Notations and Background

We consider a multi-class classification problem where given a variable $X \in \mathcal{X}$, denoting the input features of the data, we want to predict its corresponding categorical variable (i.e., class) $Y \in \{1, 2, ..., K\}$. Let $f : X \mapsto Z \in \mathbb{R}^K$ be a probabilistic model that takes input $x$ and outputs a $K$-dimensional vector $z = (z_1, z_2, ..., z_K)$ such that $\sum_{i=1}^{K} z_i = 1$. In other words, $z_k$ is the prediction probability for class $k \in \{1, ..., K\}$. Class $k$ corresponding to $\arg\max_{z_k} z$ is the predicted class.

In this work, we focus on **top-label calibration**, also commonly known as **confidence calibration**, where we are only concerned with whether or not the probability of the *predicted* class (i.e., the prediction confidence) matches with the prediction accuracy (Guo et al., 2017; Kumar et al., 2019; Kull et al., 2019). In the case of top-label calibration, the multi-class classification problem reduces to a binary classification problem, where $Y = 0$ if the class label is incorrectly predicted and $Y = 1$ if the class label is correctly predicted.

The **true calibration error (TCE)** in the setting of top-label calibration is defined as the $\ell_p$ norm of the discrepancy between the model's prediction confidence $f(X)$ and the accuracy (i.e., the true likelihood of a correct prediction) $\mathbb{E}_Y[Y|f(X)]$ (Roelofs et al., 2022):

$$\text{TCE}(f) = (\mathbb{E}_X[|f(X) - \mathbb{E}_Y[Y|f(X)]|^p])^{\frac{1}{p}} \quad (1)$$

The TCE cannot be computed analytically with a finite number of samples so a common way to estimate it is by using the *Estimated (Expected) Calibration Error* (ECE) (Roelofs et al., 2022)

$$\text{ECE}_{\mathcal{N}}(f) = \left( \frac{1}{n} \sum_{i=1}^{n} \left| f(x_i) - \frac{1}{|\mathcal{N}_i|} \sum_{j \in \mathcal{N}_i} y_j \right|^p \right)^{1/p} \quad (2)$$

where $\mathcal{N}_i$ is the neighborhood of confidence instance $i$, and the term $\frac{1}{|\mathcal{N}_i|} \sum_{j \in \mathcal{N}_i} y_j$ is the prediction accuracy in neighbourhood $\mathcal{N}_i$.

A popular approach to define the *neighborhoods* $\mathcal{N}$ is by placing the confidence values into discrete histogram bins $\mathcal{B} = \{B_1, ..., B_M\}$ such that each bin $B_i, i = 1, ..., M$ has an equal interval/width (i.e., $\frac{1}{M}$) as the other bins (Naeini et al., 2015; Guo et al., 2017), which can lead to highly biased calibration error estimates (Roelofs et al., 2022; Kumar et al., 2019; Nixon et al., 2019). The ECE$_{\text{EW}}$ calibration error estimator, based on this approach, is expressed as

$$\text{ECE}_{\text{EW}}(f) = \left( \sum_{i=1}^{M} \frac{|B_i|}{n} \left| \bar{f}(x_i) - \bar{y}_i \right|^p \right)^{1/p} \quad (3)$$

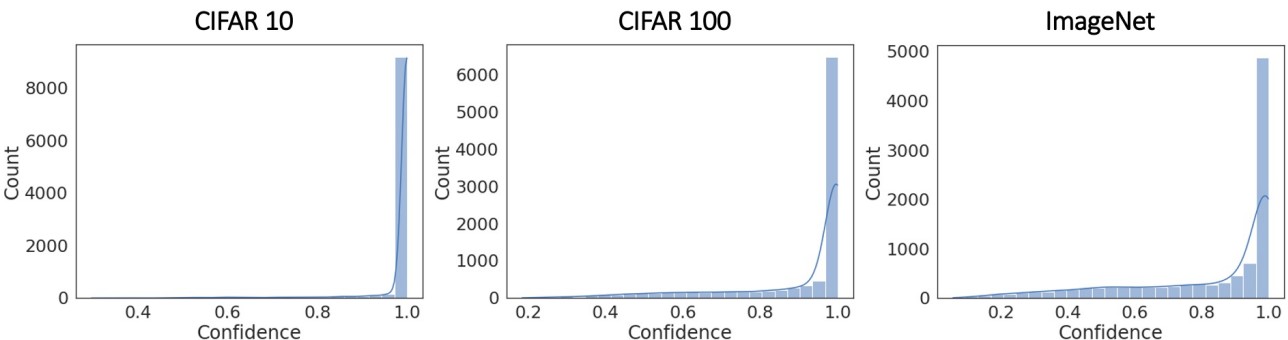

Figure 4: The CIFAR-10 dataset has the most skewed confidence distribution, the CIFAR-100 dataset is less skewed, and the ImageNet dataset is the least skewed. This figure illustrates the distribution of 10k *uncalibrated* prediction confidence values on the CIFAR-10, CIFAR-100, and ImageNet datasets. The confidence distributions shown are the prediction confidences from a ResNet-110 model for CIFAR-10/100 and a ResNet-152 model for ImageNet. Other variants of the ResNet model and Densenet models show similar confidence distributions.

Here, $n$ represents the total number of samples, $\bar{f}(x_i)$ denotes the mean prediction confidence in bin $B_i$, $|B_i|$ represents the size of bin $B_i$, and $\bar{y}_i$ indicates the prediction accuracy within bin $B_i$.

**Definition 2.1** (Bias). Bias is the difference between the ECE and the true calibration error TCE (Roelofs et al., 2022):

$$\text{Bias} = \mathbb{E}[\text{ECE}] - \text{TCE} \quad (4)$$

A *positive bias* indicates an *overestimate* of the TCE, whereas a *negative bias* indicates an *underestimate* of the TCE.

The bias can be estimated using the bias-by-construction (BBC) framework proposed by Roelofs et al. (2022), in which the TCE can be computed analytically based on an assumed confidence distribution $\mathcal{F}$ and true calibration curve $\mathbb{E}_Y[Y|f(x) = c] := T(c)$. A dataset $\{f(x_i), y_i\}_{i=1}^n$ generated from $f(x_i) \sim \mathcal{F}$ and $\mathbb{E}_Y[Y|f(x) = c] := T(c)$ is used to compute the ECE. A total of $m$ datasets is generated. The sample estimate of the bias is the difference between the mean ECE taken across $m$ simulated datasets and the TCE:

$$\widehat{\text{Bias}}(n) = \frac{1}{m} \sum_{i=1}^m \text{ECE} - \text{TCE} \quad (5)$$

## 3. KNN-based Calibration Error Estimator

In this work, we introduce an alternative calibration error estimator based on K-Nearest Neighbors (KNN). The KNN method is a commonly used non-parametric approach for density estimation due to its flexibility to adapt to any underlying probability density function and simple hyperparameter tuning (Zhao & Lai, 2020).

Closely following the ECE formulation (equation 2), we propose a KNN-based estimator (ECE$_{\text{KNN}}$) that estimates

the calibration error in the local neighborhood of each prediction confidence. ECE$_{\text{KNN}}$ is formulated as follows:

$$\text{ECE}_{\text{KNN}}(f) = \left( \sum_{i=1}^n \frac{|\mathcal{N}_i|}{\sum_{k=1}^n |\mathcal{N}_k|} \left| \bar{f}(x_i) - \bar{y}_i \right|^p \right)^{1/p} \quad (6)$$

where $\bar{f}(x_i)$ is the mean confidence of the $k$ nearest neighbors of confidence instance $i$ and $\bar{y}_i$ is the classification accuracy amongst the $k$ neighbors of $i$. Each confidence instance $i$ has a local neighborhood of size $|\mathcal{N}_i| = k$. The sum of the sizes of all local neighborhoods is $\sum_{k=1}^n |\mathcal{N}_k|$. Since there are $n$ samples (confidence instances), there is a total of $n$ local *overlapping* neighborhoods. An illustration of the KNN neighborhoods is shown in Figure 5.

While $f(x_i)$, namely the confidence value of instance $i$, is also a sensible statistic to use in place of $\bar{f}(x_i)$ in equation (6), we follow the formulation of existing estimators (ECE$_{\text{EW}}$, ECE$_{\text{EM}}$, ECE$_{\text{SWEEP}}$) and use $\bar{f}(x_i)$.

KNN neighborhoods for $z_2, z_4$ with $k = 2$

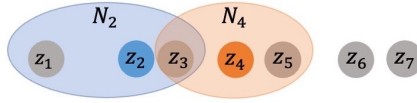

Prediction Confidences

Figure 5: An illustration of constructing overlapping K-Nearest Neighbors (KNN) neighborhoods with $k = 2$ nearest neighbors for confidence values $z_2, z_4$.

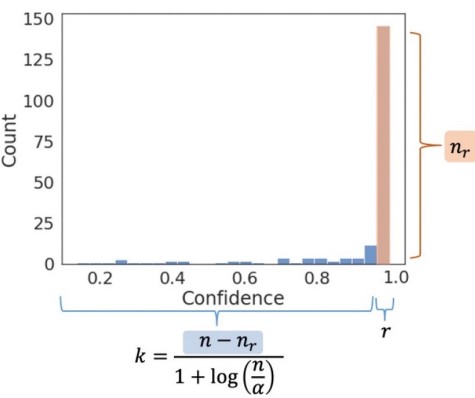

Figure 6: The process of setting of number of neighbors $k$ involves determining $n_r$, the number of samples in the region $r$ with the highest density in the confidence distribution, and setting the hyper-parameter $\alpha$ controlling the neighborhood sizes and the similarity between neighbors.

### 3.1. Choosing $k$

The key to estimating the calibration error using our proposed $\text{ECE}_{\text{KNN}}$ is constructing representative neighborhoods for each confidence instance, where a representative neighborhood should include as many samples as possible without including dissimilar neighbors.

The choice of the number of nearest neighbors ($k$) for each sample is critical, and it depends on both the *sample size* and the *distribution of the confidence values*. With a larger sample size providing a more densely distributed set of samples, a larger $k$ can be used for density estimation using KNN (Zhao & Lai, 2020) as there would be a larger number of similar neighbors. Similarly, with a larger sample size, a larger $k$ can also be used to construct more representative local neighborhoods. However, the choice of $k$ also depends on the confidence distribution, as a highly skewed distribution, such as that of the CIFAR-10 dataset (see Figure 4), may require a smaller $k$ to avoid including dissimilar neighbors in the tail end of the distribution. Conversely, a smaller $k$ may exclude many similar neighbors in a less skewed distribution, such as that of the ImageNet dataset (see Figure 4).

We propose the following three-step process for selecting the value of $k$, which is illustrated in Figure 6.

**Step 1: determine the region $r$ with the highest density in the confidence distribution** We determine the region with the highest density in the confidence distribution, $r$, separately by visual inspection for the confidence distributions of the CIFAR-10, CIFAR-100, and ImageNet datasets (see Figure 4). $r$ is smallest for the confidence distribution of CIFAR-10, with CIFAR-10 having the most skewed

distribution; $r$ is the second smallest for the confidence distribution of CIFAR-100, with CIFAR-100 having the second most skewed distribution; $r$ is the largest (relatively) for the confidence distribution of ImageNet, as ImageNet has the least skewed distribution.

**Step 2: count the number of samples in $r$** The number of samples in the most dense region $r$ is denoted as $n_r$.

**Step 3: setting $k$ based on the remaining samples** Using the remaining samples $(n - n_r)$, $k$ is set as follows

$$k \leftarrow \frac{n - n_r}{1 + \log(\frac{n}{\alpha})} \qquad (7)$$

where $n$ is the total number of samples, $n_r$ is the number of samples in the dense region $r$, and $\alpha \in (0, n]$ is a hyper-parameter that controls the relative size of the neigborhoods with respect to the sample size. Lower values of $\alpha$ result in smaller neighborhoods with higher degrees of similarity among the neighbors. In contrast, larger values of $\alpha$ lead to larger neighborhoods that can encompass a more diverse selection of neighbors.

The intuition behind Equation (7) is to adapt the neighborhood size $k$ according to the sample size $n$ for effective calibration estimation. When the sample size is small, the tail regions of the confidence distribution are sparser, necessitating larger neighborhoods $k$ relative to $n - n_r$ to ensure an adequate number of samples for reliable local CE estimation. Conversely, for larger sample sizes, the tail regions become denser, allowing us to prioritize neighborhood similarity and use a relatively smaller $k$ with respect to $n - n_r$.

To control the scaling of $k$ in a stable manner, we choose to make it inversely proportional to the logarithm of $n$, ensuring that $k$ remains larger than 1 and changes gradually as $n$ increases. The addition of $+1$ in the denominator prevents $k$ from becoming larger than $n - n_r$. The hyperparameter $\alpha$ should be chosen such that $\log(\frac{n}{\alpha})$ is greater than zero to maintain the effectiveness of the scaling.

In the subsequent section, we empirically validate the effectiveness of our proposed $\text{ECE}_{\text{KNN}}$.

## 4. Empirical Evaluation

### 4.1. Experimental Design

**Datasets** We utilize logits data[4] (Kull et al., 2019) obtained from ten different DNNs with popular backbones (e.g., ResNet, ResNet-SD, Wide-ResNet, and DenseNet), trained and evaluated on the CIFAR-10, CIFAR-100, and

---

[4]https://github.com/markus93/NN_calibration

ImageNet image classification datasets. In the uncalibrated model setting, we directly use the logits. In the recalibrated model setting, we apply Temperature Scaling[5] to recalibrate the logits.

**Baselines**   We compare the performance of our proposed KNN-based calibration error estimator ($ECE_{KNN}$) against three baseline estimators commonly used in the literature. These include the equal width ECE ($ECE_{EW}$), debiased ECE ($ECE_{DEBIASED}$), and mean sweeps ECE ($ECE_{SWEEP}$). The $ECE_{EW}$ estimator has been widely used in previous works, while the latter two, which are based on the equal mass binning scheme, have shown to exhibit lower estimation bias than other alternatives (Roelofs et al., 2022). We adopt the common practice of using 15 bins for baseline estimators ($ECE_{EW}$, $ECE_{DEBIASED}$) that employ a fixed number of bins (Park et al., 2020; Wang et al., 2020; Roelofs et al., 2022). Additionally, following Roelofs et al. (2022), we utilize the $\ell_2$ norm to measure the calibration errors.

**Bias Estimation**   We use the bias-by-construction (BBC) framework developed by Roelofs et al. (2022) to estimate the bias of the estimators on uncalibrated and recalibrated model outputs (logits) of the CIFAR-10, CIFAR-100, and ImageNet datasets across sample sizes $n = 200, 400, 800, 1600, 3200, 6400,$ and $12800$. $m = 250$ datasets are generated for each sample size to perform a sample estimate of the bias shown in equation (5). In the experiments, the TCE and ECE are computed as percentage values in $[0, 100]$, so the estimated bias is in $[-100, 100]$.

**$ECE_{KNN}$ configuration**   In Section 3, we visually inspected the confidence distributions of the CIFAR-10, CIFAR-100, and ImageNet datasets to identify the region $r$ with the highest density. We selected $r = [0.998, 1.0]$ for CIFAR-10, $r = [0.99, 1.0]$ for CIFAR-100, and $r = [0.98, 1.0]$ for ImageNet. The choice of $r$ depends on the skewness of the distributions, with smaller values for more skewed distributions and larger values for less skewed distributions. In all our experiments, we set $\alpha = 100$ to determine the value of $k$ in equation (7).

### 4.2. Empirical Results

**$ECE_{KNN}$ strikes a better balance**   Our proposed $ECE_{KNN}$ estimator successfully addresses the trade-off observed in the top-performing baseline estimators, namely $ECE_{SWEEP}$ and $ECE_{DEBIASED}$. While $ECE_{SWEEP}$ achieves lower underestimation of the calibration error (CE) on uncalibrated models, it suffers from severe overestimation on recalibrated models. Conversely, $ECE_{DEBIASED}$ reduces overestimation of the CE on recalibrated models but exhibits

---

[5]https://github.com/google-research/google-research/blob/master/caltrain/calibration_methods.py

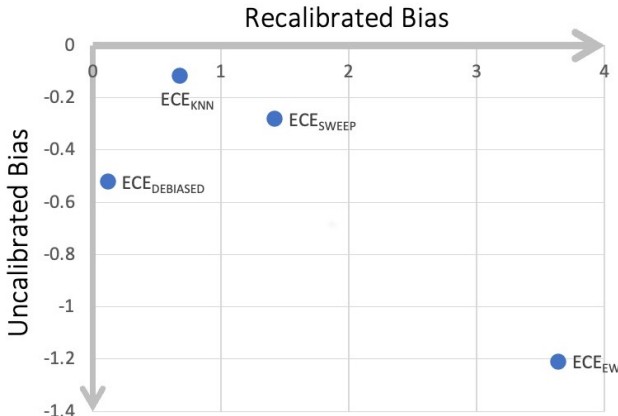

Figure 7: Comparison of calibration error estimator bias on uncalibrated and recalibrated models for CIFAR-10, CIFAR-100, and ImageNet datasets. Both the x-axis (uncalibrated bias) and y-axis (recalibrated bias) indicate better performance closer to zero. Our proposed estimator, $ECE_{KNN}$, achieves the best balance between the calibration error estimation bias on uncalibrated and recalibrated models.

substantial underestimation on uncalibrated models. This trade-off is overcome by $ECE_{KNN}$ .

Figure 7 demonstrates that our $ECE_{KNN}$ outperforms the baseline estimators in achieving a more desirable balance between estimation bias on uncalibrated and recalibrated models across diverse datasets, model backbones, and sample sizes. Notably, $ECE_{KNN}$ attains the lowest underestimation of the CE on uncalibrated models, with significantly less negative uncalibrated bias. Moreover, it effectively mitigates the degree of overestimation of the calibration error, as indicated by markedly smaller positive recalibrated bias compared to the baseline estimator with the second lowest level of underestimation on uncalibrated models, $ECE_{SWEEP}$.

**Uncalibrated models**   Our proposed $ECE_{KNN}$ surpasses the state-of-the-art metric ($ECE_{SWEEP}$) in terms of calibration error estimation bias on uncalibrated model prediction confidences across the CIFAR-10, CIFAR-100, and ImageNet datasets, as demonstrated in Figure 8. The mean absolute bias of $ECE_{KNN}$ is $0.183$ across all datasets, sample sizes, and model backbones, while $ECE_{SWEEP}$ exhibits a mean absolute bias of $0.364$. This difference in means is statistically significant ($t = 4.74$, $p < 5e - 5$).

Among all calibration metrics, $ECE_{KNN}$ achieves the lowest underestimation of the calibration error on uncalibrated models, with a mean bias of $-0.115$ across all datasets, sample sizes, and model backbones. In comparison, the mean biases of $ECE_{SWEEP}$ , $ECE_{DEBIASED}$ , and $ECE_{EW}$ are $-0.281$, $-0.521$, and $-1.210$, respectively. The difference in mean bias between $ECE_{KNN}$ and the second-best metric,

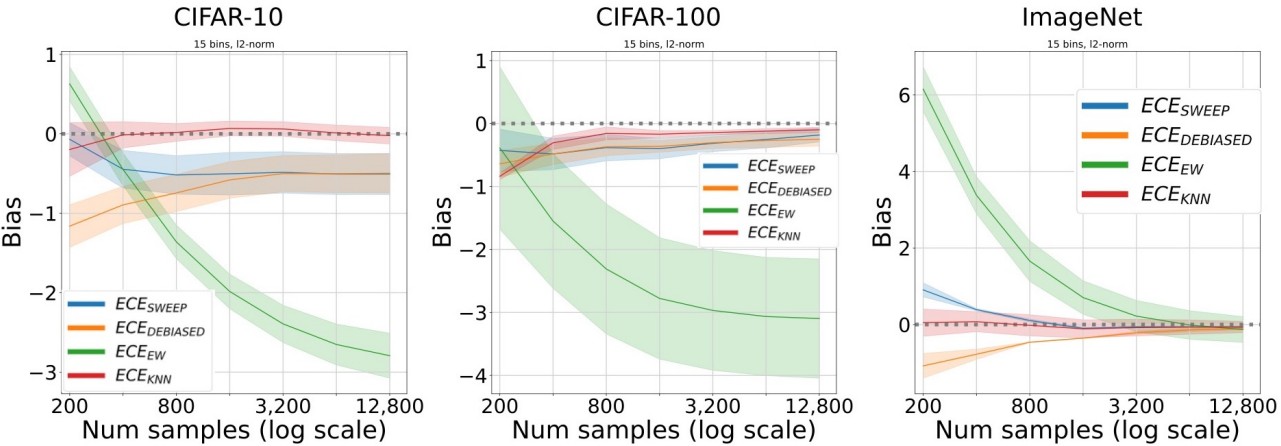

Figure 8: Calibration error estimation bias on uncalibrated models. We observe that ECE_KNN is the least biased estimator, achieving near-zero estimation bias across all datasets. These plots show the bias of calibration error estimators across different sample sizes on simulated data drawn from the confidence distribution and calibration curve fits on the CIFAR-10, CIFAR-100, and ImageNet datasets. Each line in the figure represents the mean bias across the different model backbones, with the shaded color around the line indicating the variance in bias between models.

ECE_SWEEP, is statistically significant ($t = 3.50$, $p < 5\mathrm{e}-4$).

Furthermore, when applied to the uncalibrated models of the CIFAR-10 and ImageNet datasets, ECE_KNN achieves near-zero mean bias of $-0.01$ and $-0.03$, respectively, demonstrating significantly improved accuracy in estimating the calibration error compared to all baseline estimators.

**Recalibrated models**  When applied to model prediction confidences that have been recalibrated using Temperature Scaling, our proposed ECE_KNN exhibits significantly smaller estimation bias compared to ECE_SWEEP across all three datasets, as illustrated in Figure 9. Specifically, ECE_KNN achieves a significantly lower overestimation of the calibration error on recalibrated models compared to ECE_SWEEP. Across all datasets, model backbones, and sample sizes, ECE_KNN demonstrates a mean bias of $0.676$, whereas ECE_SWEEP has a mean bias of $1.422$ ($t = 9.50$, $p \leq 5\mathrm{e} - 14$). Moreover, on recalibrated models of all three datasets, ECE_KNN achieves estimation bias comparable to that of ECE_DEBIASED when the sample size is small ($n = 200$).

## 5. Related Works

**Recalibration methods**  Several recalibration methods have been developed to improve the calibration of DNNs (Blundell et al., 2015; Gal & Ghahramani, 2015; Guo et al., 2017; Kull et al., 2019; Krishnan & Tickoo, 2020; Zhang et al., 2020; Wang et al., 2020). Deep ensemble (Lakshminarayanan et al., 2016), temperature scaling (Guo et al., 2017), and Monte Carlo dropout (Gal & Ghahramani, 2015)

are amongst the notable recalibration methods as they have shown great recalibration performance on the CIFAR-10 and ImageNet datasets in both the i.i.d. setting and under different degrees of distribution shift (Ovadia et al., 2019). Existing recalibration methods either perform *during training* or *post-hoc* calibration. *During training* calibration methods calibrate the model at training time either by estimating the distribution of model weights (Blundell et al., 2015; Louizos & Welling, 2017), training multiple models and averaging their predictions (Lakshminarayanan et al., 2016), employing additional data augmentation techniques (Thulasidasan et al., 2019), or by optimizing auxiliary loss terms that promote better calibration (Karandikar et al., 2021; Krishnan & Tickoo, 2020). *Post-hoc* calibration methods, such as Temperature Scaling, calibrate the model outputs after training by rescaling the model's logits using a single temperature parameter $T > 0$ that minimizes the model's negative log-likelihood on a held-out validation set (Guo et al., 2017). In this work, we investigate the bias in the calibration error estimation of models recalibrated using Temperature Scaling, which is widely used for its simplicity and good performance. Furthermore, Temperature Scaling is the basis of many recent recalibration methods (Park et al., 2020; Wang et al., 2020).

**Calibration metrics**  The expected calibration error (ECE_EW ) has been widely used to evaluate the calibration of DNNs and the performance of various recalibration methods (Guo et al., 2017; Ovadia et al., 2019; Park et al., 2020; Wang et al., 2020; Krishnan & Tickoo, 2020). ECE_EW estimates the discrepancy between the mean con-

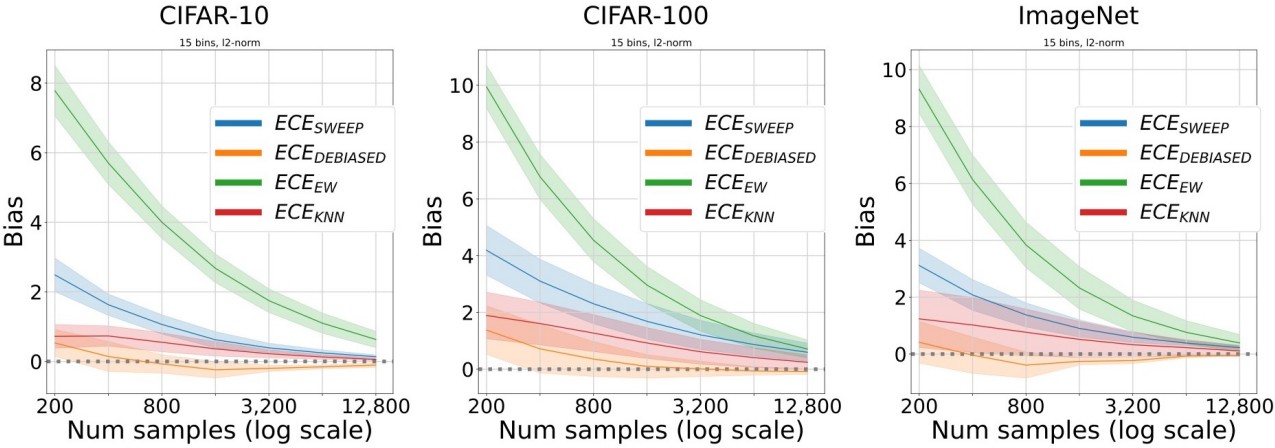

Figure 9: Calibration error estimation bias on recalibrated models. These plots show the bias of calibration error estimators across different sample sizes on simulated data drawn from the confidence distribution and calibration curve fits on the recalibrated model outputs from the CIFAR-10, CIFAR-100, and ImageNet datasets. Each line in the figure represents the mean bias across the different model backbones, with the shaded color around the line indicating the variance in bias between models.

fidence and accuracy across prediction confidences partitioned into $M$ equal-width bins (Naeini et al., 2015). However, recent studies have shown that $ECE_{EW}$ is a biased estimator of the true calibration error (TCE) and is sensitive to binning scheme and the number of bins (Nixon et al., 2019; Kumar et al., 2019; Roelofs et al., 2022). To address this, several alternative and less biased estimators have been proposed. $ECE_{EM}$ partitions the confidences into equally sized (equal mass) bins (Nixon et al., 2019), while $ECE_{DEBIASED}$ subtracts a bias term from $ECE_{EM}$ to further reduce bias (Kumar et al., 2019). $ECE_{SWEEP}$ adaptively finds the optimal number of bins based on the number of samples and the confidence distribution. Alternatively, (Zhang et al., 2020) proposes to estimate the TCE using the Kernel Density Estimator (KDE) to estimate the unknown probabilities. While the KDE-based ECE achieves the lowest estimation bias on perfectly calibrated models, it has a large bias on uncalibrated models by heavily underestimating the calibration error (Roelofs et al., 2022).

In this work, we propose an alternative estimator for the TCE based on overlapping local neighborhoods constructed using K-Nearest Neighbors. Similar to binning-based estimators, our estimator also estimates the calibration error at the local neighborhood level by computing the discrepancy between the confidence and accuracy. Unlike existing binning-based estimators, we use overlapping neighborhoods rather than disjoint ones to better capture the nuances in the calibration error at bin-edges.

## 6. Discussion

In this paper, we addressed the trade-off observed in top-performing calibration error (CE) estimators, namely $ECE_{DEBIASED}$ and $ECE_{SWEEP}$, which presents a practical challenge in medical applications. These estimators reduce underestimation of the CE on uncalibrated models at the cost of greater overestimation on recalibrated models, and vice versa. Our proposed calibration error estimator based on K-Nearest Neighbors (KNN) neighborhoods, $ECE_{KNN}$, effectively balances this trade-off.

A notable advantage of $ECE_{KNN}$ is its ability to achieve the lowest underestimation of calibration error, with near-zero estimation bias, on uncalibrated models. Additionally, it demonstrates significantly lower degrees of overestimation on recalibrated models compared to other estimators that also show relatively low underestimation on uncalibrated models ($ECE_{SWEEP}$). By employing a simple strategy to select the value of the critical hyper-parameter $k$, $ECE_{KNN}$ exhibits promising performance, highlighting its effectiveness.

In practical medical applications, where the calibration level of a given model is unknown, it is crucial for a calibration error estimator to strike a balance between the trade-off of underestimation and overestimation. Underestimating the calibration error on uncalibrated models can mislead practitioners with false certainty, potentially leading to adverse outcomes. Conversely, overestimating the calibration error on recalibrated models can result in overly conservative decision-making. Given $ECE_{KNN}$'s ability to achieve an op-

timal balance between underestimation and overestimation of the CE, it proves to be highly relevant for facilitating well-informed and reliable decision-making in healthcare settings.

**Future work**: In this study, we utilized a simple strategy to determine the value of $k$ by visually inspecting the confidence distributions and identifying the region with the highest density. Future work could focus on devising more effective strategies to optimize the value of $k$, thereby enhancing the performance of $ECE_{KNN}$. Additionally, it is important to evaluate $ECE_{KNN}$ on a wider range of datasets with different confidence distributions, as well as for models recalibrated using methods other than Temperature Scaling. Addressing these limitations would further validate the usefulness and robustness of our proposed method.

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
