# OpenReview forum: "Better Calibration Error Estimation for Reliable Uncertainty Quantification"
_ICML.cc/2023/Workshop/IMLH — IMLH 2023 Poster_

### Official Review · Reviewer_z27K · 2023-06-14
**This paper proposes a new method to quantify the calibration error based on K-nearest neighbors, which is less biased.**

**Rating:** 9
**Confidence:** 4

**Review:**

This paper identifies the biasedness of the three top-performing calibration error (CE) estimators and then proposes a less biased estimator based on KNN. This paper is well-organized, well-written and easy to understand. Based on the previous estimator that uses equal-width binning, the idea of adopting KNN is intuitive and yet effective. I just have a few suggestions/confusions.
1. In equation (6), since $|\mathcal{N}_i|=k$, is it true that $$\frac{|\mathcal{N}_i|}{\sum_{k=1}^n |\mathcal{N}_k|}$$ is always $\frac{1}{n}$?
2. Can the authors demonstrate the theoretical or intuitive explanation of equation (7)? Why do the authors choose this equation? And what does this equation mean?

---

### Official Review · Reviewer_7eir · 2023-06-18
**Novel approach to computing ECE**

**Rating:** 5
**Confidence:** 3

**Review:**

The authors propose a new approach to computing expected calibration error (ECE) of a classifier, which involves the nearest neighbors of each sample by predicted probability. They show that their metric exhibits lower bias than conventional binned ECE, as well as two previously proposed debiased versions of ECE, on three image classification datasets.

The proposed method is intuitive, and performs well empirically. The paper is well-written and easy to follow, and has clear connections to interpretability. However, the main weakness of the paper is that it has very weak connections to healthcare. The authors did not evaluate their method on any healthcare models (except Figure 3, which is not followed up later in the paper). This may be a challenging setting since healthcare data can exhibit noisy labels, and can often be highly imbalanced (and so most predicted probabilities are small). For this reason, I do not think the paper would be a great fit for the workshop.

I also have a few other comments and suggestions:
1. The authors should consider trying additional model types (e.g. different architectures and loss functions), instead of just ERM and temperature scaling, in order to better simulate real-world usage.
2. When ECE is used for model selection and comparison, arguably, the ranking of models that a metric provides is more important than the magnitude of the metric. How do each of the metrics perform when used to rank a grid of models with varying calibration levels?
3. The authors should further clarify how TCE is computed for the image datasets (presumably with this BBC framework). Why can't we just use this to compute the true calibration error instead of ECE?
4. It is not clear to me why the range of bias (e.g. in Figure 8) is so large. Since both ECE and TCE are in [0, 1], shouldn't the bias be in [-1, +1]?

---

### Official Review · Reviewer_koJ5 · 2023-06-18
**Balance the trade-off between the underestimation and overestimation**

**Rating:** 6
**Confidence:** 3

**Review:**

The authors provide a better calibration error estimator, called $ECE_{KNN}$, solving the trade-off between overestimation of the calibration error on uncalibrated models and underestimation on recalibrated models. A new design of neighbor, $i.e.$ KNN, is proposed for $eq.2$ in this paper.

Pros:
- The paper is well-organized and easy to follow.
- The method is promising and, to my knowledge, novel.

Cons:
- Since in step 1, the region should be determined based on the confidence distributions of datasets. Is there a way to know apriori what would be the optimal region, if the distribution of the dataset is unknown?
- More intuitive explanation of the process of choosing $k$ would help to understand. Can the neighbor be more dynamic?

---

### Meta-Review · Area_Chair_EnCS · 2023-06-20

**Recommendation:** Accept (Poster)
**Confidence:** 5

**Metareview:**

Uncertainty quantification is of importance for healthcare AI. The authors proposed a calibration error estimation to mitigate such problems. The paper further evaluated the biasedness of the three top-performing calibration error (CE) estimators in several benchmarks. All reviewers consensus the novelty and contribution of the paper, I suggest the authors can further discuss the comments regarding reviewers' concerns on the method and model selection. Overall, this paper has a potential contribution to the explainable AI community,  though this method is not evaluated on healthcare data, I would like to see the effects on real medical imaging datasets.

---

### Decision · Program_Chairs · 2023-06-20

Accept (Poster)